



# Investigation of the $\alpha$-pinene photooxidation by OH in the atmospheric simulation chamber SAPHIR

Michael Rolletter[1], Martin Kaminski[1,a], Ismail-Hakki Acir[1,b], Birger Bohn[1], Hans-Peter Dorn[1], Xin Li[1,c], Anna Lutz[2], Sascha Nehr[1,d], Franz Rohrer[1], Ralf Tillmann[1], Robert Wegener[1], Andreas Hofzumahaus[1], Astrid Kiendler-Scharr[1], Andreas Wahner[1], and Hendrik Fuchs[1]

[1]Institute of Energy and Climate Research, IEK-8: Troposphere, Forschungszentrum Jülich GmbH, Jülich, Germany
[2]Department of Chemistry and Molecular Biology, University of Gothenburg, Gothenburg, Sweden
[a]now at: Federal Office of Consumer Protection and Food Safety, Department 5: Method Standardisation, Reference Laboratories, Resistance to Antibiotics, Berlin, Germany
[b]now at: Institute of Nutrition and Food Sciences, Food Chemistry, University of Bonn, Bonn, Germany
[c]now at: State Key Joint Laboratory of Environmental Simulation and Pollution Control, College of Environmental Sciences and Engineering, Peking University, Beijing, China
[d]now at: INBUREX Consulting GmbH, Process Safety, Hamm, Germany

**Correspondence:** Hendrik Fuchs (h.fuchs@fz-juelich.de)

**Abstract.** The photooxidation of the most abundant monoterpene, $\alpha$-pinene, by the hydroxyl radical (OH) was investigated at atmospheric concentrations in the atmospheric simulation chamber SAPHIR. Concentrations of nitric oxide (NO) were below $120\,\mathrm{pptv}$. Yields of organic oxidation products are determined from measured time series giving values of $0.11 \pm 0.05$, $0.19 \pm 0.06$, and $0.05 \pm 0.03$ for formaldehyde, acetone, and pinonaldehyde, respectively. The pinonaldehyde yield is at the low

5   side of yields measured in previous laboratory studies, ranging from 0.06 to 0.87. These studies were mostly performed at reactant concentrations much higher than observed in the atmosphere. Time series of measured radical and trace gas concentrations are compared to results from model calculations applying the Master Chemical Mechanism (MCM) 3.3.1. The model predicts pinonaldehyde mixing ratios that are at least a factor of 4 higher than measured values. At the same time, modelled hydroxyl and hydroperoxy ($HO_2$) radical concentrations are approximately 25 % lower than measured values. Vereecken et al. (2007)

10   suggested a shift of the initial organic peroxy radical ($RO_2$) distribution towards $RO_2$ species that do not yield pinonaldehyde, but produce other organic products. Implementing these modifications reduces the model-measurement gap of pinonaldehyde by 20 % and also improves the agreement in modelled and measured radical concentrations by 10 %. However, the chemical oxidation mechanism needs further adjustment to explain observed radical and pinonaldehyde concentrations. This could be achieved by adjusting the initial $RO_2$ distribution, but could also be done by implementing alternative reaction channels of

15   $RO_2$ species that currently lead to the formation of pinonaldehyde in the model.

## 1 Introduction

Approximately 1000 Tg carbon from biogenic volatile organic compounds (BVOCs) are emitted every year into the atmosphere (Guenther et al., 2012). The majority of these compounds is isoprene (53 %) followed by monoterpene species (16 %). Within



the group of monoterpenes $\alpha$-pinene is the most abundant species with a contribution of 6.6 % to the global emission of BVOCs. During daytime, the prevalent sinks of these compounds are ozonolysis reactions and the reaction with photochemically formed hydroxyl radicals (OH) (Calogirou et al., 1999; Atkinson and Arey, 2003) producing organic peroxy radicals ($RO_2$). OH is reformed in a radical reaction chain that involves reactions with the nitric oxide (NO), thereby producing $NO_2$. This radical

reaction cycle impacts air quality, because (1) the subsequent photolysis of $NO_2$ is the only chemical source for tropospheric ozone ($O_3$) and (2) oxygenated volatile organic compounds (OVOCs) are formed, which can be precursors for the formation of secondary organic aerosols (SOA) (Glasius and Goldstein, 2016).

Field studies conducted in forested environments, which were characterised by large BVOC emissions and low NO concentrations, showed large discrepancies between measured OH radical concentrations and predictions of model calculations

(e.g. Lelieveld et al., 2008; Hofzumahaus et al., 2009; Whalley et al., 2011). Under these conditions, it is expected that radical recycling is suppressed due to the dominance of radical termination reactions such as the reaction of $RO_2$ with hydroperoxy radicals ($HO_2$). Recent theoretical and laboratory studies of the chemistry of isoprene, which was often an important OH reactant in these field experiments, however, revealed that unimolecular $RO_2$ reactions that efficiently reform OH can compete with the reaction of $RO_2$ and NO for these conditions (Peeters et al., 2009, 2014; Crounse et al., 2011, 2012; Fuchs et al.,

15 2013, 2014).

In contrast to isoprene, radical recycling in the chemistry of monoterpenes is less well investigated for conditions where field measurements indicate missing OH productions. Compared to isoprene, the degradation chemistry of monoterpene species is more complicated due to their more complex structure leading to a higher number of possible reactions and products. Laboratory and theoretical studies of the OH oxidation of $\alpha$- and $\beta$-pinene focussed on product yields and experiments were often

performed at high reactant concentrations (see for example Vereecken et al., 2007; Vereecken and Peeters, 2012; Eddingsaas et al., 2012). The main products of the $\alpha$-pinene + OH photooxidation are pinonaldehyde, acetone and formaldehyde (HCHO). Product yields determined in previous laboratory studies were highly variable. For example, pinonaldehyde yields ranged from 6 to 87 % (Larsen et al., 2001; Noziére et al., 1999).

In two field studies studies in environments in which monoterpenes and 2-methyl-3-buten-2-ol (MBO) were the most im-

portant biogenic organic compounds (Kim et al., 2013; Hens et al., 2014) missing OH production in model calculations was found. In addition, hydroperoxy radicals $HO_2$ concentrations were underestimated. If the model was constrained to measured $HO_2$, model-measurement discrepancies in OH became small due to the enhanced OH production in the reaction of $HO_2$ with NO. This indicated that the chemical system of monoterpenes as currently implemented in models lack an $HO_2$ source. A chamber study investigating the OH oxidation of $\beta$-pinene gave similar results (Kaminski et al., 2017). Another chamber

study looking at the OH oxidation of MBO showed that OH recycling in the MBO chemistry is well-understood, indicating that monoterpenes are responsible for the missing $HO_2$ in the field campaigns (Novelli et al., 2018).

In this study, the photooxidatation of $\alpha$-pinene by OH was investigated in experiments in the atmospheric simulation chamber SAPHIR (Simulation of Atmospheric PHotochemistry In a large Reaction chamber) at Forschungszentrum Jülich. Experiments were performed under controlled and atmospherically relevant conditions found in forested environments with NO mixing

ratios less than 120 pptv. Aerosol formation did not play a role in these experiments, because no significant nucleation was





observed. Measured time series were compared to model results from the Master Chemical Mechanism in the recent version 3.3.1 (MCM, 2019; Jenkin et al., 1997; Saunders et al., 2003). The impact of modifications in the chemical degradation mechanism suggested in a theoretical work by Vereecken et al. (2007) was tested. This includes the formation of new products and the change of branching ratios compared to the MCM.

## 1.1 Degradation mechanism for $\alpha$-pinene

A simplified scheme giving the reactions most relevant in the experiments here is shown in Fig. 1. The $\alpha$-pinene oxidation is initiated by the OH attack. As implemented in the MCM, OH adds to the carbon-carbon double bond of $\alpha$-pinene, forming three different $RO_2$ radicals, APINAO2, APINBO2 and APINCO2 (names and yields taken from the MCM):

$$\alpha-\text{pinene} + \text{OH} \quad \rightarrow \quad \text{APINAO2} \qquad (\text{yield} : 0.57) \qquad\qquad (\text{R1})$$

$$\alpha-\text{pinene} + \text{OH} \quad \rightarrow \quad \text{APINBO2} \qquad (\text{yield} : 0.35) \qquad\qquad (\text{R2})$$

$$\alpha-\text{pinene} + \text{OH} \quad \rightarrow \quad \text{APINCO2} \qquad (\text{yield} : 0.08) \qquad\qquad (\text{R3})$$

According to the MCM, APINAO2 and APINBO2 are the mainly produced radicals of the reaction of $\alpha$-pinene with OH with a contribution of 92 %, while APINCO2 makes only a minor contribution of 8 %. Hydrogen abstraction by OH is not considered in the MCM-mechanism. Consecutive reactions of the organic peroxy radicals with NO form alkoxy radicals, mostly APINAO and APINBO, respectively, which undergo a a fast ring-opening and subsequent $O_2$ reaction yielding pinonaldehyde and $HO_2$. This gives an overall pinonaldehyde yield of 84 % for these reaction pathways in the MCM under conditions of high NO. In contrast, the subsequent reaction of APINCO does not form pinonaldehyde, but product species include acetone and HCHO. Acetone and HCHO are also formed in the subsequent oxidation of pinonaldehyde that is significant on the time scale of our experiments. Additional reactions of peroxy radicals forming nitrates and reactions with $HO_2$ are not shown.

Vereecken et al. (2007) investigated the reaction of $\alpha$-pinene with OH using quantum-chemical calculations proposing modifications of branching ratios and additional reaction pathways. Firstly, three additional minor reaction channels of the attack of OH on $\alpha$-pinene leading to an H-abstraction are included. The total yield was calculated to be 12 %. These new pathways lead mainly to an increase in formaldehyde and also in acetone compared to the MCM. In contrast, less pinonaldehyde is formed. Secondly, the branching ratios of the other $RO_2$ species were revised. The addition of OH to the double bond was calculated to result in the OH attachment on both sites of attack forming the adducts P1OH and P2OH with similar probability. P2OH further reacts with oxygen and forms the $RO_2$ radical APINBO2. Consequently, the APINBO2 yield is increased to 44 % compared to 35 % in the MCM. The tertiary radical P1OH is chemically activated. It either is thermally stabilised forming the $RO_2$ radical APINAO2 after the $O_2$ addition, or it undergoes a prompt ring-opening of the four member ring resulting in the $RO_2$ species APINCO2. The overall yield of APINAO2 and APINCO2 is 22 % each suggested by Vereecken et al. (2007) compared to 57 % for APINAO2 and 7.5 % for APINCO2 assumed in the MCM. In addition, Vereecken et al. (2007) calculated that a 1,6-H-shift reaction and a ring-closure reaction of APINCO2 leading to 8-OOH-menthen-6-one, 2-OH-8-ooh-menthen-6-one and a dicarbonyl cycloperoxide, can compete with its reaction with NO. For conditions of the experiments





of this work the dominant pathways are both unimolecular reactions, which are faster than the reaction with NO by a factor of at least 100.

Because the formation of APINCO2 does not produce pinonaldehyde in the subsequent chemistry in contrast to APINAO2 and APINBO2, the shift in the $RO_2$ distribution in the mechanism by Vereecken et al. (2007) leads to an overall pinonaldehyde
yield of 60 % compared to 84 % in the MCM.

## 2   Methods

### 2.1   Experiments in the simulation chamber SAPHIR

The experiments in this study were performed in the outdoor atmospheric simulation chamber SAPHIR at Forschungszentrum Jülich, Germany. The chamber has been described in detail before (e.g. Rohrer et al., 2005). The cylindrical-shaped chamber
(length $18\,\mathrm{m}$, diameter $5\,\mathrm{m}$, volume $270\,\mathrm{m}^3$) is made of a double wall Teflon (FEP) film, which provides a high transmittance for the entire spectrum of solar radiation. A slight over-pressure ($30\,\mathrm{Pa}$) in the chamber ensures that no air from the outside is penetrating the chamber. A small flow of synthetic air that replaces the air sampled by the instruments and maintains the over-pressure leads to a dilution of trace gases of approximately 4 % per hour. The synthetic air used for experiments is mixed from evaporated ultrapure liquid nitrogen and oxygen (Linde, purity $\geq 99.99990\,\%$). Two fans inside the chamber are operated
to ensure a rapid mixing of trace gases. A shutter system can keep the chamber dark, for example at the beginning of an experiment, and is opened to expose the chamber air to natural sunlight to perform photooxidation experiments. Small amounts of nitric acid (HONO), formaldehyde and acetone are photolytically formed on the Teflon surface with source strengths of 100 to $200\,\mathrm{pptv}$ per hour when the chamber is illuminated by solar radiation (Rohrer et al., 2005). The primary source for OH radicals is the photolysis of HONO emitted by the chamber leading also to a continuous increase of $NO_x$ ($=NO_2+NO$) in the
experiment.

Two experiments were performed at similar conditions for this work: On 30 August, 2012 at NO mixing ratios of less than $100\,\mathrm{pptv}$ and on 02 July, 2014 at NO mixing ratios of less than $120\,\mathrm{pptv}$. Before the experiments, the chamber was flushed with synthetic air until the concentrations of trace gases from previous experiments were below the detection limits of the instruments. The chamber air was humidified by flushing water vapour from boiling Milli-Q® water into the chamber together
with a high flow of synthetic air. The relative humidity was approximately 70 % at the beginning of the experiments. $40\,\mathrm{ppbv}$ ozone produced by a discharge ozonizer (O3Onia) was injected to simulate conditions typical for forested areas before the chamber roof was opened. In the first two hours, the zero-air phase, no other reactive species were added, in order to quantify the small chamber sources for HONO, HCHO, acetone and the background OH reactivity. Afterwards, $\alpha$-pinene was injected from a high-concentration gas mixture of $\alpha$-pinene in $O_2$ prepared in a SilcoNert coated stainless steel canister (Restek) for
three times with time intervals of approximately two hours. The time between the injections allowed to study the photochemical degradation. The maximum $\alpha$-pinene concentrations were $3.8\,\mathrm{ppbv}$. After the initial phase, the OH reactivity was dominated by the injected $\alpha$-pinene and its oxidation products, so that the background reactivity becomes secondary.





The additional loss of trace gases on the chamber surface is assumed to be negligible on the time scale of the experiment. The loss of $\alpha$-pinene and pinonaldehyde, one of the oxidation products of $\alpha$-pinene, was experimentally tested by injecting $\alpha$-pinene and pinonaldehyde, respectively, into the clean chamber in the dark. The observed loss of these VOCs was consistent with the dilution due to the replenishment flow demonstrating that there was no significant loss of $\alpha$-pinene and pinonaldehyde on the Teflon film of the chamber.

## 2.2 Instrumentation

The set of instruments used in this work is listed in Table 1 giving the $1\sigma$ accuracies and precisions.

OH was measured by laser-induced fluorescence (LIF) exciting OH at 308 nm (Holland et al., 1995; Fuchs et al., 2011). Previous studies reported interferences in the OH detection by LIF for some instruments (Mao et al., 2012; Novelli et al., 2014). A laboratory study investigating potential interferences from alkene ozonolysis reactions with the LIF instrument at SAPHIR (Fuchs et al., 2016) gave no hint for significant interferences for atmospherically relevant conditions. Only for exceptionally high, non-atmospheric reactant concentrations of ozone (300-900 ppbv) and some alkenes (1-450 ppbv) interferences could be observed. Hence, no interferences are expected for conditions of the experiments in this work. In addition, OH was detected by differential optical absorption spectroscopy (DOAS, Dorn et al., 1995). OH concentration measurements of both instruments agreed on average within 15 %. A similarly good agreement between both instruments has been found in previous studies (e.g. Schlosser et al., 2009; Fuchs et al., 2012).

The LIF instrument also measured the $HO_2$ concentrations in a second fluorescence cell, in which $HO_2$ is chemically converted to OH in a reaction with added NO. Fuchs et al. (2011) reported that this detection scheme can be affected by interferences from organic peroxy radicals ($RO_2$) that also react with NO and rapidly form $HO_2$. Consequently, in the experiments of this work, the NO concentrations were reduced to suppress the conversion of $RO_2$ as described in Fuchs et al. (2011) so that interferences become unimportant.

OH reactivity ($k_{OH}$), the inverse lifetime of OH, was measured by a pump-probe instrument (Lou et al., 2010; Fuchs et al., 2017). High OH concentrations are generated in a flow-tube by laser flash photolysis of ozone in the presence of water and the decay of OH caused by ambient OH reactants is measured by LIF at the end of the flow-tube. The pseudo-first order decay rate constant fitted to the time-resolved OH measurements gives the total OH reactivity. Unfortunately, OH reactivity could only be measured in the experiment conducted in 2012, because the instrument failed in the experiment in 2014.

$\alpha$-pinene and oxygenated organic compounds expected to be formed in the $\alpha$-pinene oxidation, acetone and pinonaldehyde, were detected by a proton transfer reaction time-of-flight mass spectrometer (PTR-TOF-MS, Lindinger et al., 1998; Jordan et al., 2009). However, only for one of the experiments (02 July, 2014) the PTR-TOF-MS was calibrated to quantify pinonaldehyde. In addition, a gas chromatograph with a flame ionization detector (GC-FID) was used for the measurements of $\alpha$-pinene and acetone. VOC concentrations measured by GC-FID were on average 25 % lower than measured by PTR-MS for the experiment conducted in 2012 and 15 % lower for the experiment in 2014. This discrepancy needs to be taken into account as additional uncertainty. Formaldehyde that is also expected to be produced in the oxidation of $\alpha$-pinene was measured by a





Hantzsch monitor and by differential optical absorption spectroscopy. The measured concentrations agreed on average within 6 %.

CO and water vapour mixing ratios were monitored by a cavity ring-down instrument (Picarro), NO and $NO_2$ by a chemiluminescence instrument (Eco Physics) and $O_3$ by an UV absorption instrument (Ansyco). Photolysis frequencies were calcu-
lated from solar actinic flux densities measured by a spectroradiometer (Bohn et al., 2005; Bohn and Zilken, 2005).

## 2.3   Model calculations

The time series of trace gas compounds and radicals were calculated by a zero-dimensional box model applying the chemistry of the Master Chemical Mechanism in the recent version 3.3.1.

The MCM mechanism was extended by chamber specific processes like dilution and small sources of HONO, formaldehyde,
acetaldehyde, and acetone that are present in the sunlit chamber (Rohrer et al., 2005). Source strengths were adjusted for the individual experiments to match their production in the zero-air phase.

The dilution rate was calculated from the monitored replenishment flow rate. NO, $NO_2$, HONO, water vapour mixing ratio, temperature and pressure were constrained to measurements. While photolysis frequencies for $NO_2$, HONO, $O_3$, and pinonaldehyde were calculated from actinic flux measurement, all other photolysis frequencies were calculated for clear-sky
conditions as parameterized in the MCM 3.3.1 but scaled by the ratio of measured to calculated $j(NO_2)$ to account for cloud coverage and chamber effects. The pinonaldehyde photolysis frequency was calculated, based on the measured absorption cross sections of pinonaldehyde by Hallquist et al. (1997) and a quantum yield of one. These photolysis frequencies are greater by a factor of 3.5 compared to the parameterization in the MCM.

Modelled parameters were calculated on a one minute time base. $\alpha$-pinene and $O_3$ injections were introduced as sources
only present at the time of injection. The $O_3$ source strengths were adjusted to match measurements at the time of the injection. Similary, the $\alpha$-pinene source was adjusted to the increase of the OH reactivity. For the experiment where no $k_{OH}$ measurements were available, the increased $\alpha$-pinene concentrations measured by PTR-TOF-MS were used instead.

Sensitivity studies (M1) were performed applying modifications of the MCM based on a theoretical study by Vereecken et al. (2007). An overview of the model modifications applied to the MCM is given in Tables S2 and S3 in the supplement. The
mechanism based on Vereecken et al. (2007) differs from the MCM in new pathways, branching ratios, and product yields. $RO_2$ formed from the reactions of this modified mechanism are assumed to react similar as other $RO_2$ species with NO, $HO_2$ and other $RO_2$ species. Additional first-generation oxygenated organic compounds are formed that are not part of the MCM. In the model run denoted M1, no further reactions of these products were implemented.

Branching ratios of the initial OH + $\alpha$-pinene reaction were further adjusted in model run M2 to better match observations.
An overview of the simplified reaction scheme indicating the differences between the model runs is shown in Table 3.





## 3 Results and discussion

### 3.1 Product yields

The fate of the $RO_2$ and therefore also product yields depend on the NO concentration. At low NO levels, the reaction of $RO_2$ with $HO_2$ and $RO_2$ recombination reactions can compete with the reaction of $RO_2$ with NO. For NO mixing ratios of up to

120 ppt in the experiment on 02 July, 2014, approximately 70 % of the $RO_2$ radicals reacted with NO and 30 % with $HO_2$, while $RO_2$ self-reactions were not significant. Product species quantified in the experiments were mainly formed in the reaction of $RO_2$ with NO. In contrast, the reaction of $RO_2$ with $HO_2$ terminated the radical chain reactions and forms hydroxyperoxide species (ROOH) that were not detected.

Product yields were calculated from measured product concentrations in relation to the $\alpha$-pinene that reacted with OH. The

$\alpha$-pinene and pinonaldehyde concentrations were determined by PTR-TOF-MS. Acetone concentrations were derived from interpolated GC-FID data to exclude possible interferences on the quantifier ion of acetone in the PTR-TOF-MS. Because products were further oxidized in the experiment or had partly sources not related to the $\alpha$-pinene chemistry, a correction was applied. The correction procedure used here follows the description in Galloway et al. (2011) and Kaminski et al. (2017). The $\alpha$-pinene reacted away was corrected for dilution and its reaction with $O_3$. Ozonolysis accounted for approximately 25 %

of the loss of $\alpha$-pinene. Product concentrations were corrected for their loss from photolysis, from their reaction with OH, and from dilution. In addition, their production from the chamber sources and from the $\alpha$-pinene ozonolysis was subtracted from the measured concentrations. Acetone and HCHO chamber source strengths were determined in the initial phase of each experiment when the chamber air was already exposed to sunlight, but before the injection of $\alpha$-pinene. The productions rates were 0.04 and between 0.11 and 0.27 ppbv/h for acetone and formaldehyde, respectively. For acetone the chamber source

contributed only 10 % to the overall formed acetone. In contrast, up to 60 % of the total measured HCHO was formed on the chamber walls, which could lead to an additional bias of the determined yield. A detailed description of the corrections is given in the supplement. Yields and reaction rate constants used for the correction were taken from recommendations in Atkinson et al. (2006) also used in the MCM.

Fig. 2 shows the relation between the consumed $\alpha$-pinene and product concentrations. The product yields were determined

from the slopes of the relation resulting in yields for pinonaldehyde of $(5\pm3)$ %, for acetone of $(19\pm6)$ %, and for formaldehyde of $(11\pm5)$ %. The stated acteone and HCHO yields are the combined result from both experiments in 2012 and 2014. The stated uncertainty consists of both the measurements uncertainty and the error of the applied correction. The relationships between consumed $\alpha$-pinene and acetone and formaldehyde are not exactly linear, because both are not only directly formed in the reaction of $\alpha$-pinene with OH, but also from the subsequent oxidation of products such as pinonaldehyde. Therefore, acetone

and formaldehyde yields are increasing over the course of the experiment. The slope at the early stage of the experiment, when only little $\alpha$-pinene reacted away, reflects best their formation yield directly from the $\alpha$-pinene + OH reaction, whereas the slope at later times gives the overall yield of the $\alpha$-pinene degradation. The non-linear behaviour is most strongly pronounced for formaldehyde, for which the yield increased from 5 % to 20 % over the course of the experiment.





Results of this work are compared to results from previous studies in Table 2. Yields of pinonaldehyde that were detected by various instruments in previous studies were highly variable ranging from 6 to 87 %. High yields of 37-87 % are reported in the studies by Hatakeyama et al. (1991) and Noziére et al. (1999), in which Fourier-transform infrared spectroscopy (FT-IR) was applied. Larsen et al. (2001) found a pinonaldehyde yield of 6 % also measured using FT-IR. FT-IR measurements may suffer
from interferences from other carbonyl compounds which could have led to overestimated yields (Eddingsaas et al., 2012). Studies measuring pinonaldehyde with GC-FID (Arey et al., 1990; Hakola et al., 1994; Jaoui and Kamens, 2001; Aschmann et al., 2002; Lee et al., 2006) and PTR-MS (Lee et al., 2006; Wisthaler et al., 2001) gave similar yields that are in the range of 28 to 34 %.

Except for some experiments performed in the absence of NO, experiments in previous studies were done under conditions,
when $RO_2$ reacted mainly with NO.

In a recent study by Isaacman-VanWertz et al. (2018), the carbon budget was analysed during the photooxidation of $\alpha$-pinene by OH making use of various mass spectrometry instruments. While the carbon budget was found to be closed at the end of their experiment, the initial phase showed a discrepancy of up to 30 % between measured species including pinonaldehyde and $\alpha$-pinene that reacted away. This indicates that a substantial fraction of products was not detected, consistent with low
pinonaldehyde yields found in this and previous studies. Although no pinonaldehyde yield was reported, the yield can be estimated to be less than 20 % from figures shown in Isaacman-VanWertz et al. (2018).

The pinonaldehyde yield in the study here agrees within the stated errors with the yields reported by Larsen et al. (2001), but is lower than in all other previous studies, which used significantly higher $\alpha$-pinene mixing ratios of hundreds ppbv or even several ppmv. Concentrations of this work were close to those typically found in ambient air. This appears to be the major
difference between the experiments here and previous experiments.

The acetone yield in this study of $19 \pm 6$ % is higher by nearly a factor of two compared to previously reported values. The higher acetone yield corresponds to the lower pinonaldehyde yield and could therefore be a result of reaction pathways that do not lead to the formation of pinonaldehyde but forming acetone instead.

Only few of the previous studies reported formaldehyde yields. The yield determined in this study agrees within the stated
uncertainty with values in Noziére et al. (1999) (zero NO), Larsen et al. (2001), Wisthaler et al. (2001), Lee et al. (2006), but is significantly lower than the yield in the studies by Hatakeyama et al. (1991) and Noziére et al. (1999) (high NO). Like for acetone, additional pathways not included in the mechanism could lead to HCHO formation instead of pinonaldehyde. It is also not clear, if HCHO yields of the different studies are comparable, because the yield increases, if organic products are further oxidized during the experiment (Fig. 2). The studies by Hatakeyama et al. (1991) and Noziére et al. (1999) (high NO)
were performed at high reactant concentrations and high NO concentrations which accelerate the oxidation rate.

Few studies were performed in the presence of water, which can have an impact on product yields as shown for the product yields in the ozonolysis of $\alpha$-pinene (Tillmann et al., 2010). A water dependence in the OH degradation mechanism of $\alpha$-pinene has not been reported yet. In general, yields of products strongly depend on the fate of $RO_2$. Most of the studies were performed at high reactant concentrations and also in the presence of high NO concentrations. Therefore $RO_2$ recombination reactions
might have played a larger role compared to the chamber experiment here. In addition, the fast oxidation of $\alpha$-pinene led to





particle formation in some of the studies, and therefore, additional heterogeneous chemistry affected the results (e.g. Noziére et al., 1999). The chamber study here was performed at atmospheric reactant concentrations such that the $RO_2$ lifetime was approximately 0.5 minutes with respect to both reactions with $HO_2$ and NO, respectively, but was long enough that potential isomerization reactions could compete. The differences in the $RO_2$ fate likely explain the large variety of yields in the different

studies. This demonstrates the importance to perform experiments at atmospheric levels of reactants as done in this study.

### 3.2   Comparison of trace-gas measurements with MCM 3.3.1 model calculations

Time series of measured species are compared to model calculations using the MCM for the experiment conducted on the 02 July, 2014 (Fig. 3 and Fig. 4). This experiment is discussed here in more detail because measurements of pinonaldehyde were available in contrast to the experiment in 2012. Time series for the experiment in 2012 are shown in Fig. 5.

After the first $\alpha$-pinene injection, OH chemistry is dominated by reactions with $\alpha$-pinene. Thereby formed pinonaldehyde is overestimated in the MCM by a factor of 4. The pinonaldehyde concentrations increase directly after the VOC injections, but start to decrease one hour later due to its consumption by photolysis and reaction with OH. The model underestimates OH and $HO_2$ concentrations by approximately 25 %. Because too much pinonaldehyde is formed in the model, the OH consumption is overestimated which can partly be the reason for the smaller OH concentrations than observed.

Three $\alpha$-pinene injections with concentrations of 2-3 ppbv each were done. The modelled $\alpha$-pinene consumption is slightly slower by approximately 10 % than measured, consistent with the lower modelled OH compared to measured OH. This is also seen in an slower decrease in the modelled OH reactivity compared to measurements done in the experiment in 2012 (Fig. 5). Because OH reactivity is dominated by $\alpha$-pinene specifically shortly after its injection, this supports that $\alpha$-pinene decays are slower in the model compared to observations.

The production of acetone in the model matches the observations within the stated errors. In contrast, the formation of formaldehyde is slightly overestimated by around 10 %.

Modeled and measured $O_3$ concentrations start to slightly deviate in the second half of the experiment but agree over the whole experiment within the measurement uncertainty.

### 3.3   Sensitivity model calculations

Sensitivity model runs were performed to test, if shortcomings of the MCM model results can be explained by either recent studies reported in literature or further adjustments.

Figure 3 shows in orange colour a sensitivity run with $HO_2$ and pinonaldehyde concentrations constrained to measurements. Modeled and measured OH concentrations agree within the stated uncertainty and the time behaviour is reproduced in contrast to the MCM model run. This indicates that the radical budget is closed. As a result of the higher OH concentration the $\alpha$-

pinene is consumed faster compared to the MCM and the resulting decay reproduces the observations within the measurement uncertainty. Constraining only the $HO_2$ data is not sufficient, because the OH loss by pinonaldehyde would be overestimated.

The application of the mechanism by Vereecken et al. (2007) reduces the pinonaldehyde yield by 24 % compared to the MCM reducing the model-measurement discrepancy by a factor of 2 (Fig. 4). The lower pinonaldehyde yield results also in an





increased OH concentration, because decomposition products like acetone which reacts slower with OH is produced instead. Therefore, the OH loss rate constant is reduced compared to the results obtained with the MCM. This reduces the model-measurement discrepancy to 10 %. As discussed above, higher OH concentrations lead also to a faster consumption of the $\alpha$-pinene, so that also the model-measurement agreement of $\alpha$-pinene is improved to within 5 %. This is also consistent with

results obtained in the experiment performed in 2012, when OH reactivity was measured. OH reactivity is approximately 20 % higher than predicted by the MCM at the end of the experiment, whereas agreement within 8 % is achieved when modifications by Vereecken et al. (2007) are applied. However, there is more uncertainty in the total OH reactivity determined from the model, because reaction rate constants of the products dicarbonyl cycloperoxide, 8-OOH-menthen-6-one, and 2-OH-8-OOH-menthen-6-one, that are formed instead of pinonaldehyde from the subsequent reaction of APINCO2 can only be estimated.

Here, a similar reaction rate constant like the one for pinonaldehyde is assumed.

The modifications suggested by Vereecken et al. (2007) reduce the model-measurement discrepancies for radicals, OH reactivity, $\alpha$-pinene, and pinonaldehyde without changing the reasonable agreement for formaldehyde and acetone, but measured pinonaldehyde concentrations are still significantly lower than predicted by the model. Because APINCO2 is the only $RO_2$ species that does not form pinonaldehyde, another sensitivity study (M2) was performed for which the $RO_2$ distribution is

adjusted, such that modelled pinonaldehyde concentrations match observations. This requires a yield of APINCO2 of 86 % making the prompt ring-opening reaction subsequent of the OH attachment to $\alpha$-pinene the most important pathway. The yields for the other two $RO_2$ species are consequently reduced to 5 % and 0 % for APINAO2 and APINBO2, respectively. The minor reaction pathways suggested by Vereecken et al. (2007) (H-abstractions) remained unchanged in this model run.

If this model modification is applied, $HO_2$ radical concentrations are increased by up to 30 % giving reasonable agreement

within the stated uncertainties between modelled values and measurements (Fig. 4). The increased $HO_2$ together with a reduced OH loss rate due to the decreased pinonaldehyde concentration result in up to 30 % higher modelled OH radical concentrations compared to M1, which agree with measurements within the measurement uncertainty. In M2, acetone and formaldehyde are now underestimated by approximately 20 % and 10 %, respectively, because the production by the photooxidation of pinonaldehyde is decreased. The underestimation of acetone and formaldehyde may be caused by the missing unknown degradation

chemistry of products which are postulated in the mechanism by Vereecken et al. (2007). Vereecken et al. (2007) suggest that, for example, dicarbonyl cycloperoxide formed in the revised oxidation scheme likely produces acetone.

The change of the $RO_2$ yields is only one possibility to match the measured pinonaldehyde mixing ratios and does not imply that this is the correct oxidation scheme. If the initial $RO_2$ branching ratio suggested by Vereecken et al. (2007) is correct, then unknown reactions of APINAO2 and APINBO2 or APINAO and APINBO, respectively which suppress the pinonaldehyde

formation could also explain the discrepancies. These unknown reactions need to be significantly faster than the the currently known reactions to compete with the other reactants NO, $HO_2$ and $RO_2$.

### 3.4   Comparison with previous studies

The oxidation scheme of $\beta$-pinene that has a similar structure as $\alpha$-pinene has previously been investigated in the SAPHIR chamber for comparable conditions (Kaminski et al., 2017) giving similar results obtained for $\alpha$-pinene here. $\alpha$- and $\beta$-pinene



are isomers, which differ from each other by the position of the double bond. The double bond is endocyclic in $\alpha$-pinene and exocyclic in $\beta$-pinene. Like the assumed major oxidation product, pinonaldehyde, for $\alpha$-pinene, the main oxidation product of $\beta$-pinene, nopinone, was found to be overestimated by up to a factor 3 using the MCM model calculations in Kaminski et al. (2017). Similar to $\alpha$-pinene, Vereecken and Peeters (2012) suggested a dominant ring-opening reaction after the addition of

OH to the double bond of $\beta$-pinene that leads to other products than nopinone. The chamber study by Kaminski et al. (2017) confirms that the measured nopinone concentration is consistent with this mechanism. In addition, these reaction pathways can lead to a faster production of $HO_2$ and improves the model-measurement agreement of OH and $HO_2$ concentrations.

  Two field campaigns in environments in which monoterpene species were the dominant reactive organic compounds showed large discrepancies between modelled and observed OH and $HO_2$ radical concentrations. During the Bio-hydro-atmosphere

interactions of Energy, Aerosols, Carbon, $H_2O$, Organics, and Nitrogen–Rocky Mountain Organic Carbon Study (BEACHON-ROCS) campaign in 2010 in a forested area, the main biogenic organic compounds were 2-methyl-3-butene-2-ol (MBO) and monoterpenes with on average mixing ratios of $1.6\,\mathrm{ppbv}$ and $0.5\,\mathrm{ppbv}$, respectivly. Model calculations conducted with the University of Washington Chemical Model (UWCM) underestimated $HO_2$ radical concentrations by up to factor of 3 and OH concentrations could only be reproduced by the model, if $HO_2$ was constrained to measurements (Kim et al., 2013).

During the HUMPPA-COPEC (Hyytiälä United Measurements of Photochemistry and Particles in Air – Comprehensive Organic Precursor Emission and Concentration study) field campaign in 2010 in a boreal forest in Finland, $\alpha$-pinene mixing ratios peaked around $1\,\mathrm{ppbv}$. Again, the model calculations with CAABA/MECCA (Chemistry As A Boxmodel Application/Module Efficiently Calculating the Chemistry of the Atmosphere) gave similar results as reported by Kim et al. (2013) for the site in the Rocky Mountains. Modeled $k_{\mathrm{OH}}$ and $HO_2$ concentrations were both underestimated by model calculations. Hens et al. (2014)

attributed this to a missing $HO_2$ source.

  Results from both field campaigns are consistent with findings in the chamber experiments for $\alpha$-pinene here and $\beta$-pinene reported by Kaminski et al. (2017). MBO was also an important OH reactant during the BEACHON-ROCS campaign, but is likely not responsible for the observed model-measurement discrepancies. A chamber experiment reported by Novelli et al. (2018) demonstrated that the radical budget in the oxidation of MBO is well understood by current chemical models. In

addition, quantum-chemical calculations by Knap et al. (2016) showed that H-shift isomerization reactions are negligible for $RO_2$ radicals formed in the reaction of OH with MBO.

  In a recent laboratory study, Eddingsaas et al. (2012) investigated the $\alpha$-pinene oxidation by OH at low $NO_x$ conditions, when the fate of $RO_2$ radicals was dominated by $RO_2 + HO_2$ reactions. The authors suggest that pinonaldehyde is formed from $RO_2 + HO_2$ through an alkoxy radical channel that regenerates OH (Eddingsaas et al., 2012) and from the photooxidation and

photolysis of $\alpha$-pinene hydroxy hydroperoxides formed in this reaction. Pinonaldehyde yields were not measured but estimated to be around $33\,\%$ for low NOx. These results do not contradict results here, because only approximately $30\,\%$ of the $RO_2$ reacted with $HO_2$ in the chamber experiment and less hydroxy hydroperoxides are formed.

  Recently Xu et al. (2019) evaluated unimolecular reaction pathways in the photooxidation of $\alpha$-pinene by OH. The authors described that the hydroxy group and carbon-carbon double bond found in APINCO2 enhances the rates of unimolecular

reactions. This is consistent with the faster $HO_2$ production observed in this work that was reproduced by the model after





increasing the branching ratio of the APINCO2 formation. Xu et al. (2019) further reported that unimolecular reactions in the $\alpha$-pinene degradation do not convert NO to $NO_2$ and therefore impact the $O_3$ formation. For our experiment conditions no discrepancies in the $O_3$ concentrations are observed.

## 4   Summary and conclusions

The photooxidation of $\alpha$-pinene was investigated in the atmospheric simulation chamber SAPHIR. Two experiments were performed under atmospheric $\alpha$-pinene concentrations ($\leq 3.8\,\mathrm{ppbv}$) and medium NO conditions ($\leq 120\,\mathrm{pptv}$). Measured time series were compared to model results applying the recent version of the Master Chemical Mechanism version 3.3.1.

Model calculations lead to approximately 25 % lower OH and $HO_2$ radical concentrations than measured. In addition, pinonaldehyde is the major organic oxidation product in the MCM (84 %), whereas the measured pinonaldehyde yield is only $(5 \pm 3)$ % in the chamber experiment. This is in the lower range of previous pinonaldehyde yields determined in laboratory experiments which range between 6 % and 87 %. This large range might reflect the variety of conditions in the experiments. In addition, laboratory studies were often done at high NO and $\alpha$-pinene concentrations. The chamber study in this work is the first one using atmospheric conditions of reactant concentrations. Yields of acetone ($0.19 \pm 0.06$) and formaldehyde ($0.11 \pm 0.05$) in this study are reproduced by model calculations applying the MCM.

Reaction pathways from quantum-chemical calculations by Vereecken et al. (2007) were implemented in sensitivity model runs leading to a reduction in the model-measurement discrepancies for the radical and pinonaldehyde concentrations by approximately 10 % and 25 %, respectively. The major change is due to a shift in the branching ratio of the $RO_2$ distribution after the OH addition to $\alpha$-pinene, favouring reaction pathways that do not lead to the production of pinonaldehyde. Further adjustments of this distribution can bring model predictions into agreement with observations, but also unknown other reaction pathways that reduce the pinonaldehyde yield could explain the observations.

A chamber study on $\beta$-pinene by Kaminski et al. (2017), supported by quantum-chemical calculations by Vereecken and Peeters (2012), reported that a similar shift in the initial $RO_2$ branching ratio towards a ring-opening reaction was needed to explain product distribution and radical concentrations. Results are consistent with findings in field studies (Kim et al., 2013; Hens et al., 2014) where monoterpene emissions were high. Similar to the chamber studies for $\alpha$- and $\beta$-pinene, models gave significantly less OH and $HO_2$ compared to measured values.

Further experiments with a more detailed analysis of the organic oxidation products could help to clarify the exact reaction mechanism and further support results from quantum-chemical calculations.

*Data availability.* Data of the experiments in the SAPHIR chamber used in this work is available on the EUROCHAMP data homepage (https://data.eurochamp.org/) (Eurochamp, 2019).



*Author contributions.* MR analysed the data and wrote the paper. HF and MK designed the experiments. HF conducted the HOx radical measurements and SN was responsible for the OH reactivity measurements. BB conducted the radiation measurements. MK and RW were responsible for the GC measurements. RT, AL, and IHA were responsible for the PTR-MS measurements. XL was responsible for the HONO measurements and HPD for the DOAS OH data. FR was responsible for the NOx and $O_3$ data. All co-authors commented on the manuscript.

5 *Competing interests.* The authors declare to have no competing interests.

*Acknowledgements.* This work was supported by the EU Horizon2020 program Eurochamp2020 (grant agreement no. 730997). This project has received funding from the European Research Council (ERC) under the European Union's Horizon 2020 research and innovation program (SARLEP grant agreement no. 681529). S. Nehr and B. Bohn thank the Deutsche Forschungsgemeinschaft for funding (grant BO 1580/3-1).



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



**Table 1.** Instrumentation for radical and trace gas measurements.

| Species | Technique | Time resolution | $1\sigma$ precision | $1\sigma$ accuracy |
|---|---|---|---|---|
| OH | DOAS[a] | 205 s | $0.6 \times 10^6$ cm$^{-3}$ | 6.5 % |
| OH | LIF[b] | 47 s | $0.6 \times 10^6$ cm$^{-3}$ | 13 % |
| HO$_2$ | LIF[b] | 47 s | $1.5 \times 10^7$ cm$^{-3}$ | 16 % |
| $k_{OH}$ | Laser-photolysis + LIF[b] | 180 s | 0.3 s$^{-1}$ | 0.5 s$^{-1}$ |
| NO | Chemiluminescence | 180 s | 4 pptv | 5 % |
| NO$_2$ | Chemiluminescence | 180 s | 2 pptv | 5 % |
| O$_3$ | UV-absorption | 10 s | 1 pptv | 5 % |
| $\alpha$-pinene, pinonaldehyde & acetone | PTR-TOF-MS[c] | 40 s | 15 pptv | 14 % |
| $\alpha$-pinene & acetone | GC-FID[d] | 30 min | (4-8) % | 5 % |
| HONO | LOPAP[e] | 300 s | 1.3 pptv | 10 % |
| HCHO | DOAS[a] | 100 s | 20 % | 10 % |
| photolysis freq. | spectroradiometer | 60 s | 10 % | 10 % |

[a] DOAS = Differential Optical Absorption Spectroscopy

[b] LIF = Laser-Induced Fluorescence

[c] PTR-TOF-MS = Proton Transfer Reaction Time-Of-Flight Mass Spectrometer

[d] GC-FID = Gas Chromatography – Flame Ionization Detector

[e] LOPAP = Long-Path-Absorption-Photometer





**Table 2.** Yields of pinonaldehyde, formaldehyde and acetone for the reaction of $\alpha$-pinene + OH compared to literature values. Experimental conditions and applied measurement technique for the detection of organic compounds are additionally listed.

| Reference | Yield / % | | | Exp. conditions | | | Technique |
| --- | --- | --- | --- | --- | --- | --- | --- |
| | Pinonaldehyde | Acetone | HCHO | $\alpha$-pinene / ppbv | NO / ppbv | Water / rH % | |
| Arey et al. (1990) | 29 | - | - | 400–900 | 10000 | 0 | GC-FID |
| Hatakeyama et al. (1991) | $56 \pm 4$ | - | $54 \pm 5$ | 950–1300 | 390–2300 | 9 | FT-IR |
| Hakola et al. (1994) | $28 \pm 5$ | - | - | 350–1000 | 10000 | 0 | GC-FID |
| Noziére et al. (1999) | $87 \pm 20$ | $9 \pm 6$ | $23 \pm 9$ | 200–2700 | 4000 | 0 | FT-IR |
| Jaoui and Kamens (2001) | 28 | - | - | 940–980 | 430–490 | 18–40 | Denuder, GC-MS |
| Larsen et al. (2001) | $6 \pm 2$ | $11 \pm 3$ | $8 \pm 1$ | 1400–1600 | 1000 | 2-5 | FT-IR |
| Aschmann et al. (2002) | $28 \pm 5$ | - | - | 400–900 | 7000–9000 | 0 | GC-FID |
| Lee et al. (2006) | $30 \pm 0.3$ | 6 | 16 | 109 | 9 | 0 | PTR-MS |
| Noziére et al. (1999) | $37 \pm 7$ | $7 \pm 2$ | $8 \pm 1$ | 200–2700 | NO free | 0 | FT-IR |
| Wisthaler et al. (2001) | $34 \pm 9$ | $11 \pm 2$ | $8 \pm 1$ | 1000–1300 | NO free | 0 | PTR-MS |
| this work | $5 \pm 3$ [a] | $19 \pm 6$ [b] | $11 \pm 5$ [b] | 3.8 | <0.1 | 30–60 | PTR-MS |

[a] yield determined in the 2014 experiment

[b] combined yield from experiments in 2012 & 2014



**Table 3.** $RO_2$ yields for the different model runs and the resulting pinonaldehyde yields for the chamber experiments.

| model run | Yield / % | | | | |
|:---:|:---:|:---:|:---:|:---:|:---:|
| | APINAO2 | APINBO2 | APINCO2 | pinonaldehyde | |
| MCM | 57.2 | 35.3 | 7.5 | 84 | MCM 3.1.1 |
| M1 | 22 | 44 | 22 | 60 | Vereecken et al. (2007) |
| M2 | 5 | 0 | 83 | 5 | adjusted to measured pinonaldehyde yield |





**Figure 1.** Simplified reaction of $\alpha$-pinene as described in the MCM and modifications suggested by Vereecken et al. (2007) (shown in red). The hydrogen abstraction consists of three different pathways with a total contribution of 12 %. RO$_2$ reactions with NO forming nitrate species are not shown. See text for details. The names are taken from the MCM (black) and according to Vereecken et al. (2007) (red).



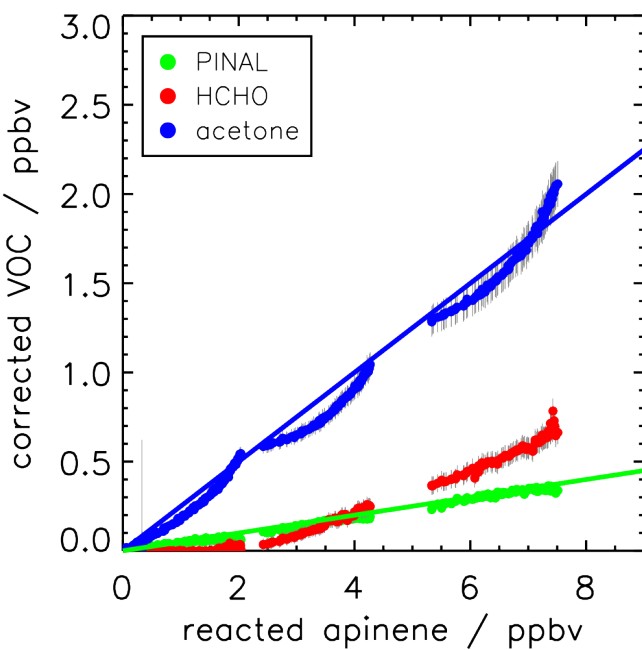

**Figure 2.** Yield of pinonaldehyde, acetone and formaldehyde determined from the slope of the relation between consumed $\alpha$-pinene and measured oxidation product concentrations for the experiment on 02 July 2014. Organic oxidation product concentrations are corrected for losses and production not related to the $\alpha$-pinene + OH reaction (see text for details). Coloured lines give the results of a linear regression. The HCHO yield is determined from the initial slope, but increases at later times of the experiments as indicated by the increasing slope of the relationship.

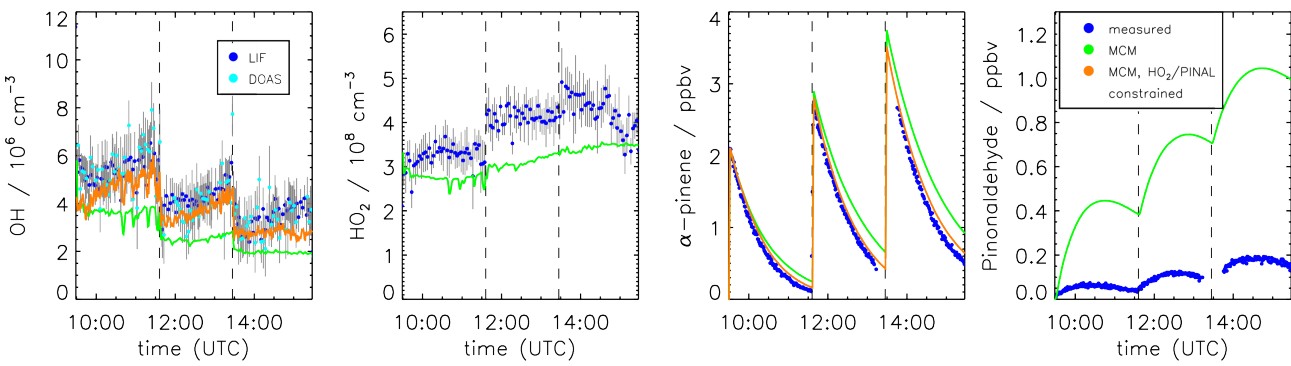

**Figure 3.** Time series of measured and modelled concentrations of radicals, $\alpha$-pinene and pinonaldehyde for an MCM model run with and without having $HO_2$ and pinonaldehyde constrained to measurements (experiment on 02 July 2014).



**Figure 4.** Time series of measured and modelled concentrations of radicals, inorganic and organic compounds during the $\alpha$-pinene photooxidation at low NO (experiment on 02 July 2014).



**Figure 5.** Time series of measured and modelled concentrations of radicals, inorganic and organic compounds during the $\alpha$-pinene photooxidation at low NO (experiment on 30 August 2012).