# Peer review of "Investigation of the $\alpha$ -pinene photooxidation by OH in the atmospheric simulation chamber SAPHIR"

_Atmospheric Chemistry and Physics, 2019_

## Referee Comment (RC1) · Anonymous Referee #1 · 9 Jul 2019

The present work describes an atmospheric simulation chamber work in SAPHIR concern on the photooxidation of the most abundant monoterpene, $\alpha$-pinene, by the hydroxyl radical (OH) at atmospheric concentrations. As a result, the organic oxidation products were found to be formaldehyde, acetone, and pinonaldehyde. However, the author found a quite different pinonaldehyde yield compared with previous study and MCM. They suggest adjusting the initial RO2 distribution to reduce the model-measurement gap of pinonaldehyde. The results are interesting and I would suggest the publication of this article with minor revision, otherwise I recommend publication as is: - Page 4 line 25, how the RH was maintained as 70% during the experiment? If not, how much it decreased when the experiment finish? - Page 5 line 31 and Table 2,

"PTR-MS" changed to "PTR-TOF-MS"
* * *

---

## Short Comment (SC1) · 29 Jul 2019

Comment on Rolletter et al. Submitted by Lu Xu, John Crounse, and Paul Wennberg

Rolletter et al. performed extensive measurements to investigate the $\alpha$-pinene photooxidation by OH under atmospherically relevant conditions. One important and interesting finding is that the measured pinonaldehyde yield is only 0.05, the lowest yield ever reported (previous measurements range from 0.06 to 0.87). Further, by comparing measurements and 0-D box model results based on different mechanisms, the authors pointed out that both Master Chemical Mechanism and theoretical study by Vereecken et al. (2007) lead to significantly higher pinonaldehyde yields.

[Figure]

Two critical parameters in determining the pinonaldehyde yield and HOx concentrations are the fraction of OH adding onto the less-substituted olefinic carbon (denoted as BROH_less_sub) and the ring-opening fraction of activated hydroxy alkyl radicals (denoted as BRring-open). In current manuscript, these two parameters are described solely on the bases of previous theoretical calculations. Undiscussed are the experimentally constrained BROH_less_sub and BRring-open for $\alpha$-pinene+OH photoxidation recently reported[1]. We suggest that BROH_less_sub is $\sim$70%, based on the OH addition branching ratio for 2-methyl 2-butene, a compound sharing similar substitutions around the C-C double bond with $\alpha$-pinene[2]. Using this constraint, we recommend BRring-open is very high (suggested to be 97%) based on the isomer distribution of $\alpha$-pinene hydroxy nitrates. We recommend that Rolletter et al. implement these experimentally-constrained values into their box model simulations and evaluate the model performance in terms of both $\alpha$-pinene oxidation products and HOx budget. We further suggest the authors rephrase their discussions on the mechanism for acetone formation. In current manuscript (Figure 1, Page 3 Line 17, etc), it is implied that acetone is directly formed from the decomposition of ring-opened alkoxy radical. Both theoretical and experimental studies[1, 3] have shown, however, that decomposition is negligible for the ring-opened alkoxy radical. The non-linear relationship between acetone and consumed $\alpha$-pinene as observed in this study provides further evidence that secondary formation of acetone is an important (perhaps the dominant) source of acetone from $\alpha$-pinene oxidation.

Reference

1. Xu, L.; Møller, K. H.; Crounse, J. D.; Otkjær, R. V.; Kjaergaard, H. G.; Wennberg, P. O. Unimolecular Reactions of Peroxy Radicals Formed in the Oxidation of a-Pinene and b-Pinene by Hydroxyl Radicals. The Journal of Physical Chemistry A 2019, 123, 1661-1674.

2. Teng, A. P.; Crounse, J. D.; Lee, L.; St. Clair, J. M.; Cohen, R. C.; Wennberg, P. O. Hydroxy Nitrate Production in the Oh-Initiated Oxidation of Alkenes. Atmos. Chem.

Phys. 2015, 15, 4297-4316.

3. Vereecken, L.; Muller, J. F.; Peeters, J. Low-Volatility Poly-Oxygenates in the Oh-Initiated Atmospheric Oxidation of ÎŚ-Pinene: Impact of Non-Traditional Peroxyl Radical Chemistry. Phys. Chem. Chem. Phys. 2007, 9, 5241-5248.

---

## Referee Comment (RC2) · Anonymous Referee #2 · 30 Jul 2019

Rolletter and coworkers studied the photoxidation of a-pinene with OH radical at atmospheric concentrations ( around 4 ppb) and under low NOx conditions (<120 ppt) at the SAPHIR chamber. They compared the results with MCM and implemented additional mechanism pathways, using those proposed by Vereecken et al. to fit the experimental results with model calculations. The main adjustement arise from the production of less pinonaldehyde that could explain changes in OH, OH2 and RO2

The work carried out and the data obtained is of high quality and I recommend it for publication with a very minor revision

Page 7 Line 16. How was the products production from the chamber source deter-

mined?. In previous experiments?. Do the follow a first order decay?

Page 7. Line 8. Did you use any specific instrumentation or techniques for trying to measure hydroperoxydes?

Does the calculated yield error correspond to 1 sigma?

---

## Author Comment (AC1) · 13 Aug 2019

The comment was uploaded in the form of a supplement:
https://www.atmos-chem-phys-discuss.net/acp-2019-492/acp-2019-492-AC1-supplement.pdf

---

## Author Response (AR1)

We would like to thank you for reviewing our manuscript and providing helpful comments.

**Anonymous Referee #1**

The present work describes an atmospheric simulation chamber work in SAPHIR concern on the photooxidation of the most abundant monoterpene, $\alpha$-pinene, by the hydroxyl radical (OH) at atmospheric concentrations. As a result, the organic oxidation products were found to be formaldehyde, acetone, and pinonaldehyde. However, the author found a quite different pinonaldehyde yield compared with previous study and MCM. They suggest adjusting the initial RO2 distribution to reduce the model-measurement gap of pinonaldehyde. The results are interesting and I would suggest the publication of this article with minor revision, otherwise I recommend publication as is:

**Comment 1:** *Page 4 line 25, how the RH was maintained as 70% during the experiment? If not, how much it decreased when the experiment finish?*
**Response:** The chamber was only humidified in the beginning of an experiment. Afterwards the relative humidity decreased mainly due to the increase of the temperature during the day to approximately 20 % in the end of an experiment.
We rephrase the sentence on page 4 line 25: "The relative humidity was approximately 70 % at the beginning of the experiments and decreased mainly due to the increase of the temperature during the day to approximately 20 % in the end of an experiment."

**Comment 2:** *Page 5 line 31 and Table 2, "PTR-MS" changed to "PTR-TOF-MS*
**Response:** This was changed as suggested.

**Anonymous Referee #2**

Rolletter and coworkers studied the photoxidation of a-pinene with OH radical at atmospheric concentrations ( around 4 ppb) and under low NOx conditions (<120 ppt) at the SAPHIR chamber. They compared the results with MCM and implemented additional mechanism pathways, using those proposed by Vereecken et al. to fit the experimental results with model calculations. The main adjustement arise from the production of less pinonaldehyde that could explain changes in OH, OH2 and RO2 The work carried out and the data obtained is of high quality and I recommend it for publication with a very minor revision

**Comment 1:** *Page 7 Line 16. How was the products production from the chamber source determined?. In previous experiments?. Do the follow a first order decay?*
**Response:** To estimate the source strengths the zero air phase, when the chamber was already exposed to sunlight but no reactants were added, was used. The source terms were parameterized as a function that depends on temperature, relative humidity and radiation as described by Rohrer et al. (2005). This function was scaled for each experiment, such that the observed acetone and HCHO timeseries were matched during the zero air phase when no chemical production was expected. It is assumed that the scaling factors remained constant over the course of one experiment. This procedure is used for the evaluation of all experiments.
We rephrase page 6 line 10: "The chamber sources were implemented as continuous sources that are parameterized by temperature, relative humidity and radiation as described by Rohrer et al. (2005). This function was scaled for each experiment, such that the observed acetone and HCHO timeseries were matched during the zero air phase when no chemical production was expected. It was assumed that the scaling factors remained constant over the course of one experiment."

**Comment 2:** *Page 7. Line 8. Did you use any specific instrumentation or techniques for trying to measure hydroperoxydes?*
**Response:** Unfortunately no measurements of hydroperoxydes were available.

**Comment 3:** *Does the calculated yield error correspond to 1 sigma?*
**Response:** The uncertainty for the pinonaldehyde yield is 1 $\sigma$ derived from measurements in one experiment in 2014, when pinonaldehyde was quantified. Yields from two experiments were determined for acetone and formaldehyde conducted in

2012 and 2014. Stated yields are the average of the two values and the error is giving the range of values derrived in the two experiments.

Page 7 line 26 was changed to: "The uncertainty for the pinonaldehyde yield is 1 $\sigma$ derived from measurements and errors of the applied correction in one experiment in 2014, when pinonaldehyde was quantified. The stated acetone and HCHO yields are the combined result from both experiments in 2012 and 2014 and the error gives the range of values derived in the two experiments."

**Comment by Lu Xu, John Crounse, and Paul Wennberg**

*Rolletter et al. performed extensive measurements to investigate the $\alpha$-pinene photooxidation by OH under atmospherically relevant conditions. One important and interesting finding is that the measured pinonaldehyde yield is only 0.05, the lowest yield ever reported (previous measurements range from 0.06 to 0.87). Further, by comparing measurements and 0-D box model results based on different mechanisms, the authors pointed out that both Master Chemical Mechanism and theoretical study by Vereecken et al. (2007) lead to significantly higher pinonaldehyde yields.*

**Comment 1:** *Two critical parameters in determining the pinonaldehyde yield and HOx concentrations are the fraction of OH adding onto the less-substituted olefinic carbon (denoted as BROH_less_sub) and the ring-opening fraction of activated hydroxy alkyl radicals (denoted as BRring-open). In current manuscript, these two parameters are described solely on the bases of previous theoretical calculations. Undiscussed are the experimentally constrained BROH_less_sub and BRring-open for $\alpha$-pinene+OH photoxidation recently reported (Xu et al., 2019). We suggest that BROH_less_sub is 70%, based on the OH addition branching ratio for 2-methyl 2-butene, a compound sharing similar substitutions around the C-C double bond with $\alpha$-pinene (Teng et al., 2015). Using this constraint, we recommend BRring-open is very high (suggested to be 97%) based on the isomer distribution of -pinene hydroxy nitrates. We recommend that Rolletter et al. implement these experimentally-constrained values into their box model simulations and evaluate the model performance in terms of both $\alpha$-pinene oxidation products and HOx budget.*

**Response:** For the sensitivity run M2 described in our manuscript BROH_less_sub is 94 % and the subsequent ring-opening is assumed to be 100 % (BRring-open). The yield of the ring-opening is similar to the values derived by Xu et al. (2019) and a shift in the $RO_2$ distribution towards APINCO2 is found in both studies. We performed an additional sensitivity run with the proposed branching ratio by Xu et al. (2019) leading to a initial $RO_2$ distribution for APINAO2/APINBO2/APINCO2 of 0.02/0.28/0.60 and 0.10 for H-abstraction reactions. The results are shown in Fig. S1.

We add on page 10 line 18: "A similar shift in the $RO_2$ distribution towards APINCO2 was proposed by Xu et al. (2019). The authors reported a branching ratio of 69 % for the initial OH addition forming P1OH and a branching ratio of 97 % for the subsequent ring-opening reaction. The resulting overall APINCO2 yield was 60 % (see Supplementary Material)."

We add the following paragraph as additional chapter S3 and Fig. S1 to the supplement: "Xu et al. (2019) studied the reaction $\alpha$-pinene + OH and proposed a mechanism constrained by experimentally determined hydroxynitrates yields. An overall shift in the initial $RO_2$ distribution towards APINCO2 was proposed. We performed an additional sensitivity run based on M1 with the proposed initial $RO_2$ distribution for APINAO2/APINBO2/APINCO2 of 0.02/0.28/0.60 and 0.10 for H-abstraction reactions. The results are shown in Fig. S1. The pinonaldehyde production is lowered by 50 % compared to M1, but is still overestimating the measured pinonaldehyde concentration by a factor of 3. The additional pinonaldehyde is derived from the higher APINBO2 fraction of 28 % compared to 5 % in M2. The formation of formaldehyde is well reproduced. In contrast, the model underpredicts the acetone production, similar to M2, but with a smaller model-measurement discrepancy of 15 %. The agreement of modeled OH and $HO_2$ concentrations is around 10 % lower compared to M2, but both agree with the measurements within the stated uncertainty."

[Figure]

**Figure S1.** Time series of measured and modelled concentrations of radicals, inorganic and organic compounds during the $\alpha$-pinene photooxidation at low NO (experiment on 02 July 2014).

**Comment 2:** *We further suggest the authors rephrase their discussions on the mechanism for acetone formation. In current manuscript (Figure 1, Page 3 Line 17, etc), it is implied that acetone is directly formed from the decomposition of ring-opened alkoxy radical. Both theoretical and experimental studies (Xu et al., 2019; Vereecken et al., 2007) have shown, however, that decomposition is negligible for the ring-opened alkoxy radical. The non-linear relationship between acetone and consumed $\alpha$-pinene as observed in this study provides further evidence that secondary formation of acetone is an important (perhaps the dominant) source of acetone from $\alpha$-pinene oxidation.*

**Response:** We rephrased our manuscript regarding the acetone production in the updated mechanism. We made it clearer in the caption of Fig. 1 that the acetone production is only relevant in the MCM and that the competing reactions in the Vereecken mechanism supress the formation of acetone.

The following sentence was added in the caption of Fig. 1: "The ring-closure and H-shift reactions in the Vereecken meachanism outrun the formation of APINCO and therefore no acetone is directly formed in contrast to the MCM"

On page 4 line 2 was added: "As a consequence, no aceton is directly formed in this pathway in contrast to the MCM mechanism."

[revised manuscript text omitted]

[a] DOAS = Differential Optical Absorption Spectroscopy

[b] LIF = Laser-Induced Fluorescence

[c] PTR-TOF-MS = Proton Transfer Reaction Time-Of-Flight Mass Spectrometer

[d] GC-FID = Gas Chromatography – Flame Ionization Detector

[e] LOPAP = Long-Path-Absorption-Photometer

**Table 2.** Yields of pinonaldehyde, formaldehyde and acetone for the reaction of $\alpha$-pinene + OH compared to literature values. Experimental conditions and applied measurement technique for the detection of organic compounds are additionally listed.

| Reference | Yield / % | | | Exp. conditions | | | Technique |
|---|---|---|---|---|---|---|---|
| | Pinonaldehyde | Acetone | HCHO | $\alpha$-pinene / ppbv | NO / ppbv | Water / rH % | |
| Arey et al. (1990) | 29 | - | - | 400–900 | 10000 | 0 | GC-FID |
| Hatakeyama et al. (1991) | $56 \pm 4$ | - | $54 \pm 5$ | 950–1300 | 390–2300 | 9 | FT-IR |
| Hakola et al. (1994) | $28 \pm 5$ | - | - | 350–1000 | 10000 | 0 | GC-FID |
| Noziére et al. (1999) | $87 \pm 20$ | $9 \pm 6$ | $23 \pm 9$ | 200–2700 | 4000 | 0 | FT-IR |
| Jaoui and Kamens (2001) | 28 | - | - | 940–980 | 430–490 | 18–40 | Denuder, GC-MS |
| Larsen et al. (2001) | $6 \pm 2$ | $11 \pm 3$ | $8 \pm 1$ | 1400–1600 | 1000 | 2-5 | FT-IR |
| Aschmann et al. (2002) | $28 \pm 5$ | - | - | 400–900 | 7000–9000 | 0 | GC-FID |
| Lee et al. (2006) | $30 \pm 0.3$ | 6 | 16 | 109 | 9 | 0 | PTR-MS |
| Noziére et al. (1999) | $37 \pm 7$ | $7 \pm 2$ | $8 \pm 1$ | 200–2700 | NO free | 0 | FT-IR |
| Wisthaler et al. (2001) | $34 \pm 9$ | $11 \pm 2$ | $8 \pm 1$ | 1000–1300 | NO free | 0 | PTR-MS |
| this work | $5 \pm 3$ [a] | $19 \pm 6$ [b] | $11 \pm 5$ [b] | 3.8 | <0.1 | 30–60 | PTR-TOF-MS |

[a] yield determined in the 2014 experiment

[b] combined yield from experiments in 2012 & 2014

**Table 3.** $RO_2$ yields for the different model runs and the resulting pinonaldehyde yields for the chamber experiments.

| model run | APINAO2 | APINBO2 | APINCO2 | pinonaldehyde | Yield / % |
|:---:|:---:|:---:|:---:|:---:|:---:|
| MCM | 57.2 | 35.3 | 7.5 | 84 | MCM 3.1.1 |
| M1 | 22 | 44 | 22 | 60 | Vereecken et al. (2007) |
| M2 | 0 | 5 | 83 | 5 | adjusted to measured pinonaldehyde yield |

[Figure]

**Figure 1.** Simplified reaction of $\alpha$-pinene as described in the MCM and modifications suggested by Vereecken et al. (2007) (shown in red). The hydrogen abstraction consists of three different pathways with a total contribution of 12 %. The ring-closure and H-shift reactions in the Vereecken meachanism outrun the formation of APINCO and therefore no acetone is directly formed in contrast to the MCM. $RO_2$ reactions with NO forming nitrate species are not shown. See text for details. The names are taken from the MCM (black) and according to Vereecken et al. (2007) (red).

[revised manuscript text omitted]

[b] value from MCM: KDEC$= 1.0 \times 10^6$ (MCM, 2019)

[c] value from MCM: KRO2HO2$= 2.91 \times 10^{-13} \exp(1300K/T)\, cm^3 s^{-1}$ (MCM, 2019)

**S3 Sensitivity study for Xu et al.**

Xu et al. (2019) studied the reaction $\alpha$-pinene + OH and proposed a mechanism constrained by experimentally determined hydroxynitrates yields. An overall shift in the initial $RO_2$ distribution towards APINCO2 was proposed. We performed an additional sensitivity run based on M1 with the proposed initial $RO_2$ distribution for APINAO2/APINBO2/APINCO2 of 0.02/0.28/0.60 and 0.10 for H-abstraction reactions. The results are shown in Fig. S2. The pinonaldehyde production is lowered by 50 % compared to M1, but is still overestimating the measured pinonaldehyde concentration by a factor of 3. The additional pinonaldehyde is derived from the higher APINBO2 fraction of 28 % compared to 5 % in M2. The formation

[Figure]

**Figure S2.** Time series of measured and modelled concentrations of radicals, inorganic and organic compounds during the α-pinene photooxidation at low NO (experiment on 02 July 2014).

of formaldehyde is well reproduced. In contrast, the model underpredicts the acetone production, similar to M2, but with a smaller model-measurement discrepancy of 15 %. The agreement of modeled OH and HO₂ concentrations is around 10 % lower compared to M2, but both agree with the measurements within the stated uncertainty